# Targeting Endometrial Cancer Stem Cell Activity with Metformin Is Inhibited by Patient-Derived Adipocyte-Secreted Factors

**DOI:** 10.3390/cancers11050653

**Published:** 2019-05-11

**Authors:** Sarah J. Kitson, Matthew Rosser, Deborah P. Fischer, Kay M. Marshall, Robert B. Clarke, Emma J. Crosbie

**Affiliations:** 1Division of Cancer Sciences, Faculty of Biology, Medicine and Health, University of Manchester, St Mary’s Hospital, Manchester M13 9WL, UK; sarah.kitson@manchester.ac.uk; 2Division of Pharmacy & Optometry, School of Health Sciences, Faculty of Biology, Medicine and Health, University of Manchester, Manchester M13 9PL, UK; matthewrosser1@gmail.com (M.R.); debbie.fischer@manchester.ac.uk (D.P.F.); kay.marshall@manchester.ac.uk (K.M.M.); 3Manchester Breast Centre, Division of Cancer Sciences, Faculty of Biology, Medicine and Health, University of Manchester, Manchester M20 4GJ, UK; 4Department of Obstetrics and Gynaecology, Manchester University NHS Foundation Trust, Manchester M13 9WL, UK

**Keywords:** Endometrial cancer, cancer stem cells, metformin, adipocytes, patient-derived

## Abstract

Advanced endometrial cancer continues to have a poor prognosis, due to limited treatment options, which may be further adversely impacted by obesity. Endometrial cancer stem cells have been reported to drive metastasis, chemotherapy resistance and disease relapse, but have yet to be fully characterised and no specific targeted therapies have been identified. Here, we describe the phenotype and genotype of aldehyde dehydrogenase high (ALDH^high^) and CD133^+ve^ endometrial cancer stem cells and how adipocyte secreted mediators block the inhibitory effect of metformin on endometrial cancer stem cell activity. Ishikawa and Hec-1a cell lines were used to characterise ALDH^high^ and CD133^+ve^ endometrial cancer cells using flow cytometry, functional sphere assays and quantitative-Polymerase Chain Reaction. The comparative effect of metformin on endometrial cancer stem cell activity and bulk tumour cell proliferation was determined using an Aldefluor and cytotoxicity assay. The impact of adipocyte secreted mediators on metformin response was established using patient-derived conditioned media. ALDH^high^ cells demonstrated greater endometrial cancer stem cell activity than CD133^+ve^ cells and had increased expression of stem cell and epithelial-mesenchymal transition genes. Treatment with 0.5–1 mM metformin reduced the proportion and activity of both endometrial cancer stem cell populations (*p* ≤ 0.05), without affecting cell viability. This effect was, however, inhibited by exposure to patient-derived adipocyte conditioned media. These results indicate a selective and specific effect of metformin on endometrial cancer stem cell activity, which is blocked by adipocyte secreted mediators. Future studies of metformin as an adjuvant therapy in endometrial cancer should be adequately powered to investigate the influence of body mass on treatment response.

## 1. Introduction

Already the fourth most common cancer in women, the incidence of endometrial cancer is rising globally as a consequence of the escalating rates of obesity, to which up to 41% of endometrial cancers can be directly attributed [1,2,3]. Whilst the majority of women present early with postmenopausal bleeding and disease confined to the uterus, there is a marked discrepancy in five-year survival between those with stage I disease (95%) and those with stage IV disease (14%) [4]. The presence of obesity may also further adversely impact prognosis [5]. Adipose tissue appears to play a direct role in metastasis, through the release of pro-inflammatory mediators (e.g., IL-6, TNF-α) which promote cancer cell migration and invasion [6], the infiltration of adipocytes into tumours and promotion of neo-vascularisation [7,8] and hormonal manipulation, in particular the release of pro-metastatic leptin, oestrogen and insulin-like growth factor-1 [9,10,11], although the relative importance of each of these pathways in endometrial carcinogenesis remains to be determined.

Treatment options for the extra-uterine disease are limited, with carboplatin and paclitaxel chemotherapy frequently used despite there being no evidence to suggest that it improves overall survival [12]. Indeed, any modest short term benefit from chemotherapy is counteracted by an ongoing risk of disease relapse after treatment is discontinued [13]. This has been hypothesised to be due to the persistence of a sub-population of tumour cells referred to as cancer stem cells [14]. These progenitor stem cells have acquired the capacity for self-renewal through the accumulation of mutations in the Wnt/β-catenin, BMI1 and Hippo signalling pathways and are able to differentiate into multiple different cell types [15,16,17,18]. They form tumours when transplanted into animal models at much lower concentrations than bulk tumour cells, have increased invasion and migration capacity and are resistant to commonly used chemotherapeutic drugs, making them responsible for metastasis and disease relapse [19].

In order to study the unique characteristics of these cells in more detail, two broad means of identifying and separating these cells from the bulk tumour population are frequently employed; detection of cell surface markers and enzyme activity, and the use of functional assays. Expression of CD133, a glycosylated pentaspan membrane protein, has been shown to be restricted to undifferentiated cells, and a population of CD133^+ve^ cells has been described in primary endometrial tumours that demonstrate many of the characteristics of cancer stem cells, including self-renewal, multipotency and increased tumour formation in immunodeficient mice [20,21,22,23]. Similar traits are exhibited by endometrial cancer cells with high activity levels of the aldehyde dehydrogenase (ALDH) enzyme, which plays a critical role in retinoic acid signalling, including increased colony and tumour forming capacity, invasive potential and resistance to the cytotoxic effects of cisplatin chemotherapy [24,25,26]. Despite this, the relationship between CD133^+ve^ and ALDH^high^ endometrial cancer cells and their relative cancer stem cell activity is unknown. Functional assays for the determination of stem cell characterisation include the formation of three-dimensional ‘spheres’ under non-adherent conditions, which would normally be associated with cell aniokis [27]. This technique has been well described in other tumour types, but has not previously been undertaken using this standardised protocol in endometrial cancer.

Regardless of the increasing awareness of the importance of cancer stem cells in disease relapse and metastasis, there have been relatively few specific therapeutic options examined to target these cells in endometrial cancer. Limited cell line data suggests ruxolitinib and nifuroxazide (JAK1 and STAT3 inhibitors, respectively) may be of benefit, but these findings have not been replicated in humans, and whilst short-term progestin treatment appears promising, effects are seen in only a proportion of women [25,28].

Evidence from other tumour types suggests that metformin may selectively target cancer stem cells by reducing sphere and in vivo tumour formation, for example, when tested in a mouse xenograft breast cancer model [29]. Indeed, the drug appears to act synergistically with other chemotherapeutic drugs, including platinum agents and doxycycline, and is associated with prolongation of disease remission [30,31]. Metformin decreases the expression of cancer stem cell related genes associated with pluripotency, self-renewal and the Hippo pathway, including SOX9, β-catenin and YAP1, and appears to reduce the proportion of ALDH^high^ ovarian cancer cells and CD133^+ve^ pancreatic cancer cells at a concentration that does not affect the proliferation of whole tumour cell lines [31,32,33,34,35]. Despite widespread interest in metformin in other aspects of the management of endometrial cancer, the effect of the drug on endometrial cancer stem cells has not previously been studied, although there is limited observational data to suggest that it may act as a chemosensitizer [36,37,38]. Whilst our own randomised controlled pre-surgical window study did not demonstrate reduced endometrial cancer cell proliferation in response to metformin exposure, as measured by immunohistochemical expression of Ki-67, this does not preclude a specific effect of the drug on endometrial cancer stem cells [39].

This study, therefore, aimed to further characterise endometrial cancer stem cells using functional assays, markers and gene expression, and to investigate the effect of metformin on these cells in vitro. It also sought to determine whether obesity could influence endometrial cancer biology by examining the effect of adipocyte secreted mediators on endometrial cancer stem cells and their response to metformin treatment.

## 2. Results

### 2.1. ALDH^high^ and CD133^+ve^ Endometrial Cancer Cells Demonstrate Cancer Stem Cell Activity and Increased Mitochondrial Mass

Both Ishikawa and Hec-1a endometrial cancer cell lines contained a small proportion of cells with the ability to survive, proliferate and form three-dimensional spherical structures in non-adherent culture. Spheres formed by the Ishikawa cell line were compact and truly spherical, whilst Hec-1a cells formed spheres with a more irregular outline (Figure 1a). The self-renewal capacity of these spheres was demonstrated by serial passaging (to at least passage 2) and increased with each passage, suggesting positive selection for cancer stem cells (data not shown).

A small proportion of Ishikawa (0.4%) and Hec-1a (3.4%) cells were found to have high ALDH activity, forming more spheres under attachment-free conditions than ALDH^low^ cells (Figure 1b). ALDH activity was thus confirmed as a marker enriching for sphere-forming activity, although ALDH^low^ cells also produced sphere colonies. 

CD133 expression also enriched for sphere formation efficiency (Figure 1c), but only in the Ishikawa cell line, where 16.8% of cells were CD133^+ve^. The Hec-1a cell line contained no CD133^+ve^ cells.

Ishikawa and Hec-1a cancer stem cells, identified by ALDH^high^ activity, had a 1.5–2.3-fold higher mitochondrial mass, as measured by MitoTracker mean fluorescent intensity (MFI) than bulk tumour cells with low ALDH activity (*p* ≤ 0.05, Figure 1d). Similarly, Ishikawa cancer stem cells expressing CD133 had greater mitochondrial mass than CD133^-ve^ cells (1.3-fold increase, *p* ≤ 0.001, Figure 1d), suggesting they may be more sensitive to mitochondrial inhibitors, such as metformin, than bulk tumour cells.

We determined the extent of overlap between the two populations of cells with cancer stem cell activity in the Ishikawa cell line using dual staining and flow cytometry. Double positive cells had the greatest sphere formation efficiency, with double negative cells forming the fewest number of spheres (Figure 1e). ALDH activity correlated better with cancer stem cell activity than CD133. The markers identified two almost exclusive populations of cells with cancer stem cell activity, with only 0.01% of cells expressing both markers (Appendix A). This was confirmed when the relative expression of epithelial and mesenchymal markers was examined in the two cell populations (Figure 1f). ALDH^high^ cells had increased expression of genes associated with both an epithelial-like and mesenchymal-like state, whilst CD133^+ve^ cells demonstrated a reduction in epithelial genes, including E-cadherin, and a corresponding increase in the mesenchymal marker vimentin (both *p* ≤ 0.001).

### 2.2. ALDH^high^ Cells Express Genes Associated with Pluripotency, Self-Renewal and a Cancer Stem Cell Phenotype, Whilst CD133^+ve^ Cells Do Not

We used RT-qPCR to determine whether cells with high ALDH activity or expressing CD133 did, indeed, express key genes associated with pluripotency, self-renewal and a cancer stem cell phenotype. Cells with high ALDH activity in both the Ishikawa and Hec-1a cell lines had increased expression of SOX2 compared with ALDH^low^ cells (both *p* < 0.05, Figure 2a). SOX2 expression was 57 times higher in ALDH^high^ cells than ALDH^low^ cells in the Ishikawa cell line (*p* = 0.02). Expression of NANOG was also increased in ALDH^high^ cells, but only in the Ishikawa cell line (*p* < 0.01). Expression of BMI1, HEY1, and HES1 increased in cells with high ALDH activity in the Ishikawa cell line (all *p* < 0.05), whilst expression of only HEY1 was increased in ALDH^high^ cells in the Hec-1a cell line (Appendix A).

In contrast, expression of SOX2 and NANOG was significantly lower in CD133^+ve^ cells compared with CD133^-ve^ cells (Figure 2a), consistent with earlier findings that ALDH activity may be a better marker of endometrial cancer stem cell activity.

As activation of the Wnt and Hippo signalling pathways is associated with the development of cancer stem cell traits, we determined the relative expression of markers of these pathways in endometrial cancer cells with high ALDH activity and CD133 expression. ALDH^high^ Ishikawa cells exhibited upregulation of the Wnt pathway genes WNT2 and CTNNB1 (Figure 2b). ALDH^high^ Hec-1a cells had increased expression of the CTNNB1 gene only. There was evidence of overactivity of Hippo signalling in the ALDH^high^ cells of both cell lines, with a 57.5 ± 13.7 (*p* = 0.01) and 3.5 ± 0.6 (*p* = 0.01)-fold increase in YAP1 expression in Ishikawa and Hec-1a cell lines, respectively. These findings were not replicated in CD133^+ve^ cells; Wnt pathway activity was lower in CD133^+ve^ cells than CD133^-ve^ Ishikawa cells (fold change WNT2 0.07 ± 0.00, *p* ≤ 0.0001, CTNNB1 0.67 ± 0.04, *p* = 0.001, Figure 2b). Expression of the Hippo target gene YAP1 was marginally increased in CD133^+ve^ cells (fold change 1.3 ± 0.03, *p* = 0.003).

We then studied the expression of transcription factors controlling epithelial-mesenchymal transition and, hence, cancer metastasis. Expression of SNAI1 was increased in ALDH^high^ cells in both cell lines studied, with Ishikawa ALDH^high^ cells also demonstrating upregulation of TWIST (Figure 2c). Expression of ZEB1 was increased 191.1 ± 48.4-fold in Hec-1a ALDH^high^ cells (*p* = 0.02, Appendix A). Expression of all three EMT transcription factors was the same or lower in CD133^+ve^ cells compared with CD133^-ve^ cells (Figure 2c, Appendix A).

### 2.3. Metformin Reduces the Number and Activity of Endometrial Cancer Stem Cells

We next sought to investigate the effect of metformin on endometrial cancer stem cell activity and number in the two cell lines. Metformin had a dose-dependent effect on cancer stem cell activity, as measured by sphere formation efficiency, in both Ishikawa and Hec-1a cell lines (Figure 3a). An effect on sphere formation efficiency was observed with as low as 0.5 mM metformin, with a 1.3-fold decrease in Ishikawa sphere formation efficiency (SFE) (*p* = 0.03) and a 1.6-fold decrease in Hec-1a SFE (*p* = 0.008). At a concentration of 20mM, almost no spheres were formed in either cell line.

Using the two lowest concentrations of metformin at which an effect on sphere formation efficiency was observed, the effect of the drug on the proportion of ALDH^high^ and CD133^+ve^ cells was determined. Metformin, at a concentration of 1 mM, significantly reduced the proportion of Hec-1a cancer stems identified by high ALDH activity by 1.7-fold (*p* ≤ 0.05, Figure 3b). By contrast, the same dose of metformin had no effect on the proportion of ALDH^high^ Ishikawa cells, although the population of ALDH^high^ cells in this cell line was markedly smaller. Treatment with metformin at 0.5 mM and 1 mM concentration resulted in a significant reduction in the number of CD133^+ve^ endometrial cancer stem cells in a dose-dependent manner (Figure 3b). The proportion of CD133^+ve^ cells was 10-fold lower after 1 mM metformin treatment (*p* ≤ 0.0001).

Whilst metformin decreased the cancer stem cell activity of the two endometrial cancer cell lines, it had no effect on the self-renewal capacity of these cells (Appendix A).

The reduction in sphere formation efficiency of the Ishikawa and Hec-1a cells was due to an effect of metformin on cancer stem cell activity rather than because increased cell death, as demonstrated in aSulforhodamine B (SRB) assay (Appendix A). A concentration of at least 1mM metformin was required to significantly reduce the viability of Ishikawa cells and whilst 0.5 mM metformin resulted in cell death in a proportion of Hec-1a cells, the effect was less than that observed on sphere formation efficiency. The IC_50_ values for metformin to inhibit cancer stem cell activity and viability in the Ishikawa cell lines were 1.60 mM and 6.77 mM, respectively, and in the Hec-1a cell line were 1.46 mM and 3.72 mM, respectively. 

### 2.4. Metformin Reduced the Expression of Cancer Stem Cell Genes

Metformin reduced the expression of genes associated with pluripotency and self-renewal in a dose-dependent manner in the Ishikawa cells, though had a less pronounced effect in the Hec-1a cells (Figure 4a, Appendix A). Metformin treatment of Ishikawa cells also resulted in a significant reduction in Wnt pathway activation and Hippo signalling in a dose-dependent manner (Figure 4b). Hec-1a cells treated with 1mM metformin had lower CTNNB1 and YAP1 expression than controls, but an increase in WNT2 expression. 

The effect of metformin on the expression of EMT transcription factors was more heterogeneous. Treatment with 1mM metformin resulted in a two-fold reduction in TWIST and SNAI1 expression (both *p* ≤ 0.05) in Ishikawa cells, whilst having no significant effect in Hec-1a cells (Figure 4c). In contrast, the same concentration increased the expression of ZEB1 in Ishikawa cells by 1.3-fold, but reduced expression in Hec-1a cells (fold change 0.9±0.03, *p* = 0.05, Appendix A).

### 2.5. Patient-Derived Adipocyte Conditioned Media Increased Endometrial Cancer Stem Cell Activity but Reduced the Sensitivity of These Cells to Metformin Treatment without Affecting Cell Proliferation

In order to investigate the impact of obesity on endometrial cancer cell proliferation and cancer stem cell activity, conditioned media was obtained from patient-derived adipocytes that had undergone maturation in response to exposure to adipocyte differentiation media. The baseline characteristics of the six women from whom pre-adipocytes were cultured are detailed in Appendix A. Pre-adipocytes treated for 15 days with differentiation media showed 100% differentiation into mature adipocytes, with a rounded morphology and positive triglyceride staining with Oil Red O. Treatment for 10 and 12 days resulted in a lesser degree of differentiation in a small proportion of cells only (Figure 5a). Pre-adipocytes exposed to growth media only retained their fibroblast-like phenotype; they did not contain lipid droplets or demonstrate positive staining with Oil Red O (Appendix A).

We determined the effect of adipocyte conditioned media on endometrial cancer proliferation and response to metformin treatment using an SRB assay. Exposure to patient-derived adipocyte conditioned media did not affect the proliferation of endometrial cancer cells grown in monolayer, neither did it alter the responsiveness of these cells to treatment with metformin (Figure 5b). Treatment with pre-adipocyte conditioned media similarly had no effect on the viability of Ishikawa cells (Appendix A).

In contrast, exposure to patient-derived adipocyte conditioned media had a significant effect on the activity of endometrial cancer stem cells. A significant increase in the cancer stem cell activity of Ishikawa cells exposed to patient-derived conditioned media from four out of six adipocyte cultures was observed (Figure 5c). On average, SFE was increased by 3.76 ± 0.81-fold compared with cells treated with control (unconditioned growth) media (*p* = 0.02). An increase in cancer stem cell activity was also noted after exposure to three out of the six corresponding pre-adipocyte conditioned medias, however, the overall effect was smaller in magnitude than that observed following treatment with mature adipocyte conditioned media (Appendix A).

In addition, the normal inhibitory effect of metformin on endometrial cancer stem cell activity was lost when these cells were pre-treated with adipocyte conditioned media for 96 h (Figure 5d). Rather than metformin treatment reducing the number of spheres formed in non-adherent culture, the SFE of Ishikawa cells was unchanged if they had been previously exposed to adipocyte conditioned media. These results suggest that under these experimental conditions, adipocyte conditioned media has a selective effect on the activity of endometrial cancer stem cells and does not affect the sensitivity of bulk tumour cells to low concentrations of metformin.

## 3. Discussion

High ALDH activity and CD133 positivity identifies two almost-mutually exclusive populations of cells in the two endometrial cancer cell lines, Ishikawa and Hec-1a, which have increased cancer stem cell activity, as determined by their ability to form spheres in non-adherent culture. ALDH^high^ cells have a greater cancer stem cell activity and have increased expression of stem cell and EMT-transcription factor genes compared with CD133^+ve^ cells. The increased mitochondrial mass of endometrial cancer stem cells makes them more susceptible to inhibitors of mitochondrial function than bulk tumour cells. Indeed, metformin reduces the number and activity of endometrial cancer stem cells at a concentration lower than that required to affect cell proliferation. It also decreases the expression of genes associated with self-renewal, pluripotency and induction of EMT, although only in the Ishikawa cell line. Adipocyte conditioned media selectively increases cancer stem cell activity, but also reduces the sensitivity of these cells to the effects of metformin, suggesting potential resistance to its anti-tumour effect in obese women.

The proportions of ALDH^high^ and CD133^+ve^ cells in the two endometrial cancer cell lines reported here are similar to those in other studies [40,41,42]. This is the first study, however, in which the two populations of cells with cancer stem cell activity have been directly compared at a functional and genetic level. Ding et al. investigated stem cell gene expression in CD133^+ve^ cells, noting that they had significantly increased expression of *SOX2* and *NANOG*, as well as *EPCAM* [22]. This contrasts with the findings described here, where RNA expression of all stem cell and epithelial markers were significantly lower in CD133^+ve^ compared with CD133^-ve^ cells. This may be because of differences between endometrial cancer cell lines and primary endometrial tumours and that the findings of Ding et al. were based on a solitary low grade, early stage endometrial cancer. Only one study has previously sought to compare the gene expression of ALDH^high^ and ALDH^low^ endometrial cancer cells and limited their investigation to EMT genes [25]. They found a difference only in Twist 1 and 2 expression between the two cell populations.

Whilst in breast cancer, distinct populations of cancer stem cells appear to exist in either an epithelial or mesenchymal-like state [43], this does not appear to be true of endometrial cancer. ALDH^high^ cells in the Ishikawa and Hec-1a cell line had increased expression of both epithelial and mesenchymal markers, with the opposite generally found in CD133^+ve^ cells. Whilst a strong link between EMT and gain of cancer stem-like cell properties has been proven, complete EMT is unusual in the majority of cancers, with most expressing both epithelial and mesenchymal markers [44,45]. Indeed, the balance between EMT and its reverse process, mesenchymal-epithelial transition, appears to be important in not only driving metastasis, but also establishing new tumours at distant sites. It is, therefore, not surprising that endometrial cancer stem cells have increased expression of both epithelial and mesenchymal markers, as well as the transcription factors that control this process. Low dose metformin, at least in endometrial cancer cell lines, appears to reduce the expression of *SNAI1*, *TWIST* and *ZEB1* and could, therefore, influence the metastatic process, although this has yet to be proven in primary endometrial cancer samples [46].

The effect of metformin on the number and activity of endometrial cancer stem cells is consistent with previous findings in breast, prostate and ovarian cancer, where it has been shown to decrease mammosphere formation and reduce tumour growth in xenograft models [29,30,31]. That this effect is at least attenuated, if not lost completely, when cells are pre-treated with adipocyte conditioned media is an important finding. Based on these results it would appear that this phenomenon is related to secreted mediators (adipokines) from mature adipocytes. Future work is needed to identify these proteins, through the use of adipokine multiplex ELISA arrays, and to develop targeted therapies against them if metformin is to be used effectively in the adjuvant setting for endometrial cancer.

An active role for adipocytes in endometrial cancer biology is becoming established. A recent study by Sahoo et al. found that adipocyte conditioned media increased the colony forming capacity of Ishikawa cells and reduced their responsiveness to paclitaxel when co-treated for 72 h [47]. In contrast to the findings of this study and our previously published clinical trial, however, the authors also described an increase in endometrial cancer cell proliferation with adipocyte conditioned media treatment. There are notable differences in methodology between the two laboratory studies, though, including the fact that Sahoo et al. used conditioned media generated from a mouse pre-adipocyte cell line that had been induced to undergo differentiation rather than primary adipocytes, and that they used a cell viability assay that detected ATP production rather than the number of cells present. Adipocyte conditioned media has been shown to increase the rate of oxidative phosphorylation in ID8 ovarian cancer cells and this could thus explain the increase in ATP production rather than a true increase in cell number [48]. The SRB utilised in this study, in contrast, relies on the binding of SRB to proteins to determine cell mass and hence avoids the potential pitfalls associated with assays which measure the metabolic activity of cells [49].

The strengths of this work include the replication of experimental results in two distinct endometrial cancer cell lines, thereby increasing the generalisability of the results obtained. The effect of metformin on both the number and function of cancer stem cells in vitro was investigated and the development of a model of obesity-associated endometrial cancer also provided an opportunity to further understand the potential mechanisms through which excess adiposity drives endometrial carcinogenesis and possibly metformin resistance.

Cell lines do, however, have their limitations, including the risk of introducing genotypic and phenotypic variation through serial passaging, which can take them away from their cell of origin [50]. They also have limited intra- and inter-tumoural heterogeneity. Replication of experiments using primary endometrial cancer cells would help to negate these issues. This work focussed solely on endometrial cancer stem cells identified through high ALDH activity and CD133 expression on the basis of previously published, albeit limited, work in the stem cell field and does not exclude the presence of other populations of endometrial cancer stem cells. The ‘gold standard’ method of assessing cancer stem cell activity, in vivo transplantation in mouse xenograft models, was not employed in this current study, but would be required to fully confirm our findings from in vitro assays of cancer stem activity. The concentration of metformin at which an effect on endometrial cancer stem cell activity was observed was lower than that used in other experiments in endometrial cancer, but was several fold higher than that measured in plasma in humans or in endometrial cancer tissue, where peak levels have not been found to exceed 200 µM [51,52,53]. Whilst, therefore, the use of higher doses of metformin may overcome the inhibitory effects of adipocyte conditioned media on the response of endometrial cancer stem cells to the drug, the clinical relevance of such findings is uncertain. The differential responsiveness of cells in vivo and in vitro to metformin may be related to the experimental conditions employed, in particular the use of high glucose containing growth media, which has been shown to alter the responsiveness of cells to metformin treatment [54].

An attempt has been made to correlate the findings described here with the response to metformin treatment in patients. Using endometrial biopsies taken as part of the PREMIUM window study, the effect of short-term (1–5 weeks) treatment with metformin or a matched placebo on the immunohistochemical expression of ALDH and CD133 was determined [39]. Blinded scoring failed to reveal any effect of metformin on the expression of these endometrial cancer stem cell markers (data not shown), however, concerns were raised about the specificity of the antibodies used and the extent of background staining. There is also an issue about the value of using immunohistochemistry in this instance, given that the antibodies employed recognize specific protein epitopes without, in the case of ALDH, being able to determine enzyme activity or, in the case of CD133, detecting antigens with different glycosylation sites. Fluorescence activated cell sorting (FACS)of fresh endometrial cancer cells would negate these problems.

Future clinical trials of metformin should focus on its efficacy as a potential maintenance treatment to reduce the risk of endometrial cancer recurrence. An initial step could be to undertake a pre-surgical window study to determine the effect of metformin on the proportion of ALDH^high^ and CD133^+ve^ primary cells detected by flow cytometry, as well as endometrial cancer sphere formation in women at high risk of disease relapse. It should be sufficiently powered to allow for the impact of obesity on treatment response to be determined. Confirmation of an effect could pave the way for longer-term treatment with the drug for up to five years with relapse-free and overall survival as outcome measures.

## 4. Materials and Methods

### 4.1. Endometrial Cancer Cell Lines

Ishikawa and Hec-1a, well established endometrioid endometrial cancer cell lines, were used for experiments (obtained from HPV Culture Collection, Salisbury, UK and ATCC, Middlesex, UK, respectively) [55]. Cells were cultured in growth media composed of DMEM/F12 (Gibco, Paisley, UK), with a glucose concentration of 4.5g/L, supplemented with 10% (*v*/*v*) foetal bovine serum (FBS, Sigma-Aldrich, Dorset, UK) and additional glutamine (1% (*v*/*v*) Gibco, Paisley, UK).

### 4.2. Adipocyte culture

To investigate the impact of obesity on endometrial cancer biology, endometrial cancer cell lines were exposed to conditioned media from primary pre- and mature adipocytes.

Pre-adipocytes were extracted from omental biopsies obtained at the time of surgery for endometrial cancer using collagenase/dispase (Roche, Sussex, UK) digestion and grown in pre-adipocyte growth media (DMEM/F12 plus 10% (*v*/*v*) FBS, 1% (*v*/*v*) glutamine, 1% (*v*/*v*) penicillin/streptomycin (Sigma-Aldrich, Dorset, UK)) in a humidified incubator at 37 °C and 5% CO2. Ethics approval was obtained from the Research Ethics Committee North West-Haydock (14/NW/1236) and all patients gave written, informed consent to participate.

Confluent pre-adipocytes were treated with adipocyte differentiation media (Sigma-Aldrich, Dorset, UK) for up to 15 days. Change in cell morphology from an elongated to a rounded shape and the detection of intracellular triglycerides and lipids with Oil Red O staining confirmed maturation [56].

### 4.3. Conditioned Media

Media was removed from confluent pre-adipocytes and mature adipocytes and replaced with phenol-red free DMEM/F12 supplemented with 1% (*v*/*v*) glutamine and 1% (*v*/*v*) penicillin/streptomycin, without the addition of FBS. This conditioned media was removed 24 h later and either used immediately or stored at −80 °C until required.

### 4.4. Sphere Formation and Passaging

Sphere formation was performed as previously described [27]. Single cells were seeded onto low adherent plates containing stem cell media (Gibco, Paisley, UK) with or without additional metformin (0.1–20 mM) and cultured for five days. The number of spheres formed was counted using a light microscope. To assess self-renewal, spheres were disaggregated and replated, with the number of secondary spheres counted after a further five days of culture. 

### 4.5. Flow Cytometry

Flow cytometry was performed using the BD LSR II flow cytometer (BD Biosciences, Oxford, UK) and analysed using FlowJo software. Cell sorting was performed using the Aria II/III flow cytometer (BD Biosciences, Oxford, UK).

### 4.6. CD133

Single dissociated Ishikawa and Hec-1a cells were suspended in buffer composed of 5% (*v*/*v*) bovine serum albumin (BSA), 2mM ethylenediaminetetracetic acid (EDTA) and PBS and incubated with either CD133-APC or CD133-VioBright FITC antibodies (both monoclonal mouse, Miltenyi Biotec, dilution 1:500) and FcR blocking reagent (Miltenyi Biotec, dilution 1:5) or isotype control antibody (mouse IgG1, Miltenyi Biotec, dilution 1:11) for 10 min in the dark at 2–8 °C. Following incubation, cells were washed and resuspended in the buffer for flow cytometry.

### 4.7. ALDH Activity

Single dissociated cells were suspended in Aldefluor assay buffer (STEMCELL technologies, Cambridge UK) and incubated with an ALDH substrate, bodipyaminoacetaldehyde (BAAA), at 1.5 mM for 45 min at 37 °C. In order to distinguish between high and low ALDH activity, a fraction of cells was incubated under identical conditions in the presence of a two-fold molar excess of ALDH inhibitor (diethylaminobenzaldehyde, DEAB). 7-actinoaminomycin-D (7AAD) was used for dead cell exclusion.

### 4.8. Mitochondrial Mass

Cancer stem cells in other cancer types have been characterised by high mitochondrial mass [57]. Single dissociated cells were thus incubated with 25 nM Mitotracker (Fisher Scientific, Loughborough, UK) dissolved in DMSO for 45 min at 37 °C either at the same time as the ALDEFLUOR reagent or prior to the addition of the CD133 antibody. 

### 4.9. Sulforhodamine B (SRB) Cytotoxicity Assay

Cell viability was determined using an SRB cytotoxicity assay (Sigma-Aldrich, Dorset, UK) in a 96-well plate with protein absorption measured at 450 nM using a microplate photometer.

### 4.10. Real Time Quantitative-Reverse Transcriptase-Polymerase Chain Reaction (RT-qPCR)

Experiments were conducted using unsorted Ishikawa and Hec-1a cells and sorted cancer stem cells according to CD133 expression and ALDH activity, which had been maintained in normal growth media with or without supplemental metformin. RNA extraction was performed using the RNeasy Micro Kit (Qiagen, Manchester, UK) according to the manufacturer’s protocol. 

Reverse transcription of 1.5ng of RNA was performed with the Reverse Transcription Mastermix (PN 100-6297, Fluidigm, San Francisco, USA). The cDNA formed was immediately pre-amplified using Taqman gene expression arrays (Appendix A, Fisher Scientific, Loughborough, UK) and Taqman PreAmp Mastermix (Fisher Scientific, Loughborough, UK). 

RT-qPCR was performed using the Flex Six™ Integrated Fluidic Circuit (IFC, Fluidigm, San Francisco, CA, USA). Pre-amplified cDNA was added to a sample pre-mix composed of Taqman Fast Advanced MasterMix (2×, PN 4444557, Fisher Scientific, Loughborough, UK) and 20× GE Sample Reagent (PN 100-6311, Fluidigm, San Francisco, CA, USA). Assays and sample solutions were pipetted into the designated IFC inlets. RT-qPCR was performed using the BiomarkHD system, with data analysed using the Biomark Real-Time PCR Analysis Software.

### 4.11. Statistical Analysis

All experiments were performed in triplicate on at least three occasions and results are presented as the mean ± standard error of the mean (SEM). The student’s *t* test or analysis of covariance was used to compare normally distributed, paired data. A two-sided *p* value ≤0.05 was considered statistically significant, with asterisk used to indicate significant results as * *p* ≤ 0.05, ** *p* ≤ 0.01, *** *p* ≤ 0.001, **** *p* ≤ 0.0001. All analyses were performed using GraphPad Prism v.7 and SPSS v.23.

Please refer to supplemental methods for further details.

## 5. Conclusions

ALDH activity and CD133 expression identify two distinct populations of endometrial cancer cells with different cancer stem cell activity and expression of stem cell and EMT genes. Metformin, at a concentration lower than that required to affect proliferation, reduces the number and activity of these cells in vitro. This effect is diminished, however, in the presence of adipocyte conditioned media. Identification of the adipokines responsible for metformin resistance should be pursued and the efficacy of long-term drug treatment on endometrial cancer stem cell activity, relapse and survival should be determined in well-designed randomised controlled trials.

## Figures and Tables

**Figure 1 cancers-11-00653-f001:**
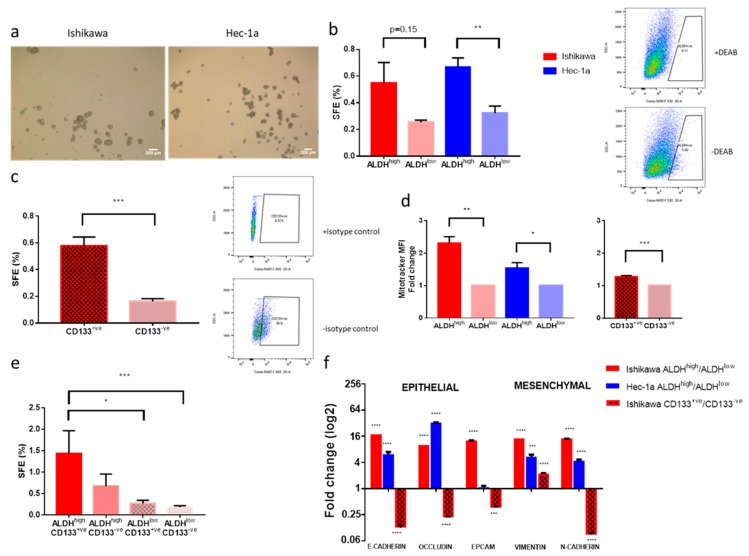
Characterisation of endometrial cancer stem cells in two cell lines. (**a**) images at ×40 magnification of ‘spheres’ formed under non-adherent conditions by two endometrial cancer cell lines. (**b**) On the left, sphere formation efficiency (SFE) of Ishikawa and Hec-1a cells with high and low aldehyde dehydrogenase (ALDH) activity (*n* = 3). On the right, a representative example of flow cytometry and gating for ALDH^high^ cells using diethylaminobenzaldehyde (DEAB), an ALDH inhibitor. (**c**) On the left, SFE of CD133^+ve^ and CD133^-ve^ Ishikawa cells (*n* = 3). On the right, a representative example of flow cytometry and gating for CD133^+ve^ cells using an isotype control antibody. (**d**) On the left, mitochondrial mass of Ishikawa and Hec-1a cells with high and low ALDH activity and, on the right, mitochondrial mass of CD133 positive and negative Ishikawa cells (*n* = 3). SFE of Ishikawa cells dual stained for ALDH activity and CD133 positivity (*n* = 4). (**e**) SFE of Ishikawa cells dual stained for ALDH activity and CD133 positivity (*n* = 4). (**f**) qRT-PCR of genes associated with an epithelial and mesenchymal phenotype in ALDH^high^ and CD133^+ve^ cells (*n* = 3). Data are represented as means ± SEM. * *p* ≤ 0.05, ** *p* ≤ 0.01, *** *p* ≤ 0.001.

**Figure 2 cancers-11-00653-f002:**
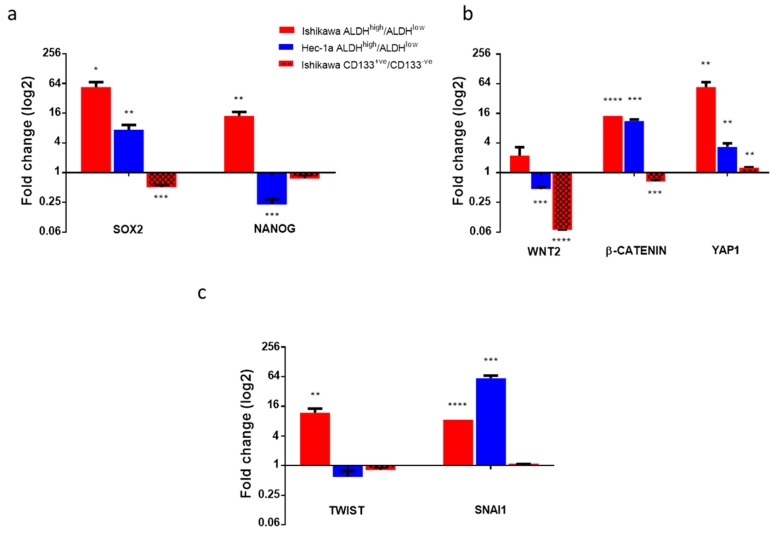
Expression of genes associated with a cancer stem cell phenotype in ALDH^high^ and CD133^+ve^ endometrial cancer cells. (**a**) qRT-PCR of genes associated with pluripotency (cells capable of giving rise to a number of different cell types) and self-renewal in ALDH^high^ Ishikawa and Hec-1a cells and CD133^+ve^ Ishikawa cells compared with ALDH^low^ and CD133^-ve^ cells, respectively (*n* = 3). (**b**) qRT-PCR of genes within the Wnt and Hippo signalling pathways in ALDH^high^ and CD133^+ve^ cells (*n* = 3). (**c**) qRT-PCR of epithelial-mesenchymal transition (EMT) transcription factors in ALDH^high^ and CD133^+ve^ cells (*n* = 3). Data are represented as mean ±SEM. * *p* ≤ 0.05, ** *p* ≤ 0.01, *** *p* ≤ 0.001, **** *p* ≤ 0.0001.

**Figure 3 cancers-11-00653-f003:**
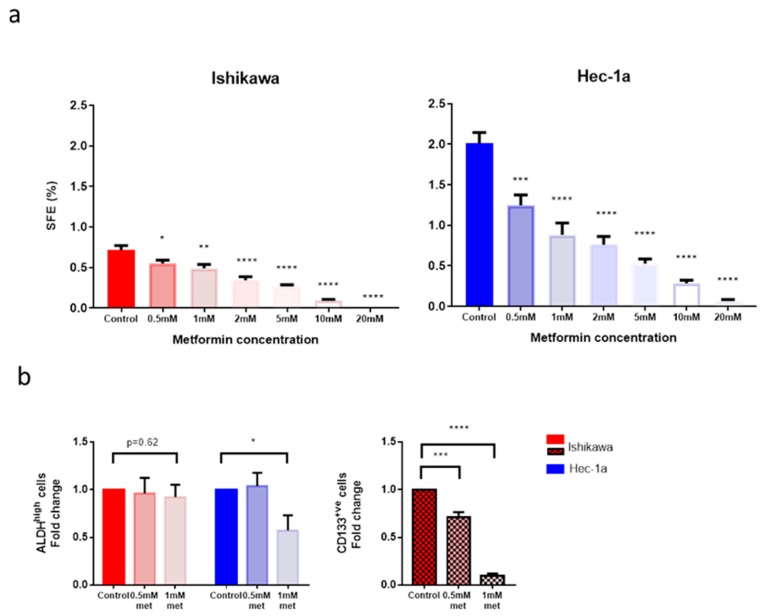
Effect of metformin on endometrial cancer stem cell number and activity. (**a**) On the left, SFE of Ishikawa cells treated with metformin in non-adherent culture. On the right, SFE of Hec-1a cells treated with metformin (*n* = 3). (**b**) On the left, the proportion of ALDH^high^ cells following treatment of Ishikawa and Hec-1a cells with metformin. On the right, the proportion of CD133^+ve^ Ishikawa cells following treatment with metformin (*n* = 5). Data are represented as mean ± SEM. * *p* ≤ 0.05, ** *p* ≤ 0.01, ** *p* ≤ 0.001, **** *p* ≤ 0.0001.

**Figure 4 cancers-11-00653-f004:**
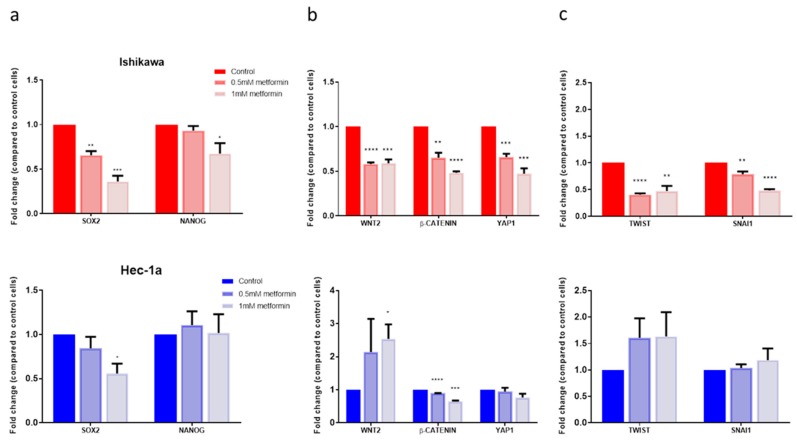
Effect of metformin on stem cell and EMT gene expression. (**a**) On the top, qRT-PCR of EMT transcription factor genes in Ishikawa cells treated with metformin. Underneath, qRT-PCR of the same genes in Hec-1a cells treated with metformin (*n* = 3). (**b**) On the top, qRT-PCR of genes associated with pluripotency and self-renewal in Ishikawa cells treated with metformin. Underneath, qRT-PCR of the same genes in Hec-1a cells treated with metformin (*n* = 3). (**c**) On the top, qRT-PCR of genes within the Wnt and Hippo signalling pathways in Ishikawa cells treated with metformin. Underneath, qRT-PCR of the same genes in Hec-1a cells treated with metformin (*n* = 3). Data are represented as the mean ± SEM. * *p* ≤ 0.05, ** *p* ≤ 0.01, *** *p* ≤ 0.001, **** *p* ≤ 0.0001.

**Figure 5 cancers-11-00653-f005:**
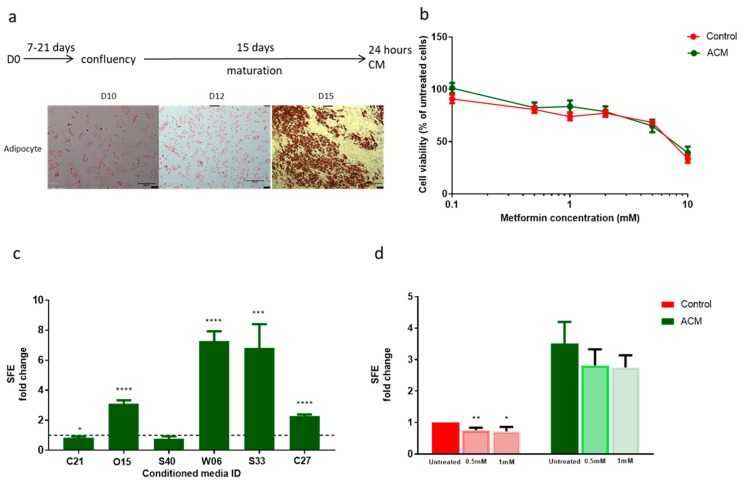
Effect of pre-adipocyte and adipocyte conditioned media (ACM) on endometrial cancer stem cell activity and response to metformin. (**a**) Schematic of adipocyte maturation and conditioned media generation and images at ×100 magnification of pre-adipocytes and mature adipocytes stained with Oil Red O after 10, 12 and 15 days of exposure to adipocyte differentiation media. (**b**) SRB cytotoxicity assay of Ishikawa cells exposed to ACM and treated with metformin (*n* = 4). (**c**) SFE of Ishikawa cells treated with individual patient adipocyte conditioned media (ACM, *n* = 6). (**d**) SFE of Ishikawa cells exposed to ACM and metformin (*n* = 6). Data are represented as mean ± SEM from individual patient ACM. * *p* ≤ 0.05, ** *p* ≤ 0.01, *** *p* ≤ 0.001, **** *p* ≤ 0.0001.

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
