# Peer review of "Targeting Endometrial Cancer Stem Cell Activity with Metformin Is Inhibited by Patient-Derived Adipocyte-Secreted Factors"

_cancers, 2019, doi:10.3390/cancers11050653_

Round 1

Reviewer 1 Report

Manuscript ID: cancers-463244

Title: Targeting endometrial cancer stem cell activity with metformin is inhibited by patient-derived adipocyte-secreted factors

Authors: Sarah Kitson, Matthew Rosser, Deborah Fischer, Kay Marshall, Robert Clarke, Emma Crosbie *

Comments

In order to evaluate the effect of adipocytes on endometrial cancer stem cells, firstly the authors isolated small portion of cells, which possessed self-renewal capacity, from Ishikawa and Hec-1a cell lines. They found that aldehyde dehydrogenase (ALDH) high and/or CD133-positive cells contained more mitochondrial mass compared with ALDH-low or CD133-negative cells. These cancer stem cells showed mesenchymal phenotype markers. ALDH-high cells, however, expressed key genes relating to pluripotency, self-renewal, and epithelial-mesenchymal transition more than CD133-positive cells. Then the authors showed the effect of metformin on cancer stem cell from Ishikawa and Hec-1a cell lines. Moreover, they found that adipocyte conditioned media had selective inhibitory effect only on cancer tumor cells. In patients data, metformin treatment was not associated with the reduction of ALDH-high or CD133-positive cells. 

Although the manuscript provide informative data, there are several points to be discussed:

If possible, add more information of the patients providing pre-adipocytes during operation — age or staging.

Add more information on ALDH and CD133-staining patient profiles.

Author Response

Comment 1- If possible, add more information of the patients providing pre-adipocytes during operation — age or staging.

Response 1- We thank the reviewer for their comment. The characteristics of women from whom omental biopsies were obtained for primary pre-adipocyte culture and maturation are detailed in supplementary table S2, including age, BMI, grade and stage of endometrial cancer.

Comment 2- Add more information on ALDH and CD133-staining patient profiles.

Response 2- The characteristics of women in the PREMIUM study who contributed tumour for ALDH and CD133 immunohistochemical analysis are detailed in supplementary figure S6, including age, BMI, grade and stage of endometrial cancer.

Reviewer 2 Report

This manuscript by Kitson et.al characterizes a ALDH high and CD133+ endometrial cancer cells  (Ishikawa and Hec-1a cell lines) for CSC activity.  They find ALDH+ cells are most stem-like.  They then evaluate the impact of metformin on CSC activity and find that metformin reduces the percentage of ALDH+ cells.  However, they find that adipose cell secreted factors prevents the anti-CSC activity of the metformin. They thus suggest that adiposity should be considered in clinical trials evaluating metformin in patients.

The paper has several strengths including:

·         This is the first study of metformin on endometrial CSC (to my knowledge – though metformin has been studied targeting CSC in other cancer types)

·         They use physiologically relevant metformin doses.

·         They evaluate the impact of host cells (adipocytes) on tumor response.

·         They evaluate the impact in patient specimens

The paper has several weaknesses as well including a lack of novelty in the CSC studies --ALDH, CD133 (and CD44) have all been reported as CSC markers in endometrial ca in several publications. Below are major concerns that I have

1.       To characterize the CSC population they perform standard assays – FACSs, EMT/stem cell mRNA expression signatures, Sphere forming capacity, but they do not perform the gold standard which is limiting dilution tumor initiation capacity.  While time consuming, this is essential and should really be done to confirm the work.

2.       The impact of metformin on endometrial CSC is new (though this has been reported in other cancers including ovarian cancer). They find a dose dependent decrease reduction in spheroid formation.  The results are a bit incongruous with their conclusions as they conclude ALDH is the best CSC maker, but metformin does not impact ALDH+ Ishikawa cells (though it does impact CD133+ cells).  This should be discussed.

3.       The ability of adipocyte conditioned media to overcome the benefits of metformin are interesting as well.  However, it would be nice to see if higher (and still physiologically obtainable) doses of metformin (2mm, 6mm) could overcome some of the effects of the ACM.

4.       The analysis of the premium study is important however, the methods on how the study was done are lacking—which antibodies were used? Which ALDH isoform was tested?  Most IHC test ALDH1A1 but recent studies suggest ALDH1A3 (and occasionally ALDH1A2) are major contributors to ALDEFLUOR activity.  If ALDH1A1 was used then ALDH1A3 should also be evaluated.  If live frozen cells are available Aldefluor/CD133 FACS analysis would be preferable as this well evaluate ALDH activity regardless of isoform.

Minor issues that would help:

                It would be nice to see ALDH and CD133 FACS together to see how many A+C+ cells there are.

Author Response

Reviewer #2

Comment 1- To characterize the CSC population they perform standard assays – FACSs, EMT/stem cell mRNA expression signatures, Sphere forming capacity, but they do not perform the gold standard which is limiting dilution tumor initiation capacity.  While time consuming, this is essential and should really be done to confirm the work.

Response 1- We thank the reviewer for their comment. While we have performed many confirmatory stem cell activity assays we agree with the reviewer that validation using the ‘gold standard’ limiting dilution tumour initiation assay would be an essential confirmation of the data. However, due to time constraints it was not possible to perform this work for this revision. We, therefore, highlight in the discussion on page 10 that we would seek to address this in future experiments. The text in the discussion reads: “…The ‘gold standard’ method of assessing cancer stem cell activity, in vivo transplantation in mouse xenograft models, was not employed in this current study, but would be required to fully confirm our findings from in vitro assays of cancer stem activity…”

Comment 2- The impact of metformin on endometrial CSC is new (though this has been reported in other cancers including ovarian cancer). They find a dose dependent decrease reduction in spheroid formation.  The results are a bit incongruous with their conclusions as they conclude ALDH is the best CSC maker, but metformin does not impact ALDH+ Ishikawa cells (though it does impact CD133+ cells).  This should be discussed.

Response 2- We thank the reviewer for their insightful comment. Although treatment with 1mM metformin did result in a reduction in the proportion of ALDHhigh Ishikawa cells, we believe that the absence of a statistically significant effect was due to the low number of these cells within the cell line. We discuss this in the manuscript on page 6: “…By contrast, the same dose of metformin had no effect on the proportion of ALDHhigh Ishikawa cells, although the population of ALDHhigh cells in this cell line was markedly smaller…”

Comment 3- The ability of adipocyte conditioned media to overcome the benefits of metformin are interesting as well.  However, it would be nice to see if higher (and still physiologically obtainable) doses of metformin (2mm, 6mm) could overcome some of the effects of the ACM.

Response 3- It is good to see that the reviewer found our results of interest. Whilst the inhibitory effect of ACM on the response of endometrial cancer stem cells to metformin may be overcome with higher doses of the drug, analysis of serum metformin levels from women recruited into the PREMIUM revealed that mean trough levels are 7µM (±0.74µM) (1) and peak levels do not exceed 200µM (2). The clinical relevance of using higher doses of metformin in vitro is not supported and so we have not tested these higher doses. The discussion has been amended to include this point on page 10-11 “…The concentration of metformin at which an effect on endometrial cancer stem cell activity was observed was lower than that used in other experiments in endometrial cancer, but was several fold higher than that measured in plasma in humans or in endometrial cancer tissue, where peak levels have not been found to exceed 200µM [51-53]. Whilst, therefore, the use of higher doses of metformin may overcome the inhibitory effects of adipocyte conditioned media on the response of endometrial cancer stem cells to the drug, the clinical relevance of such findings is uncertain. The differential responsiveness of cells in vivo and in vitro to metformin may be related to the experimental conditions employed, in particular the use of high glucose containing growth media, which has been shown to alter the responsiveness of cells to metformin treatment [54]…”

Comment 4- The analysis of the premium study is important however, the methods on how the study was done are lacking—which antibodies were used? Which ALDH isoform was tested?  Most IHC test ALDH1A1 but recent studies suggest ALDH1A3 (and occasionally ALDH1A2) are major contributors to ALDEFLUOR activity.  If ALDH1A1 was used then ALDH1A3 should also be evaluated.  If live frozen cells are available Aldefluor/CD133 FACS analysis would be preferable as this well evaluate ALDH activity regardless of isoform

Response 4- Further details about the experimental conditions employed during the immunohistochemistry are provided in the supplementary methods. The antibody used was raised against ALDH1 and will, therefore, detect both ALDH1A1 and ALDH1A3. This has now been clarified in the manuscript. Unfortunately live frozen cells are not available to be able to undertake FACS analysis, although this has been highlighted in the discussion as an area to be addressed in future work “…Future clinical trials of metformin should focus on its efficacy as a potential maintenance treatment to reduce the risk of endometrial cancer recurrence. An initial step could be to undertake a pre-surgical window study to determine the effect of metformin on the proportion of ALDHhigh and CD133+ve primary cells detected by flow cytometry as well as endometrial cancer sphere formation in women at high risk of disease relapse…”

Comment 5- It would be nice to see ALDH and CD133 FACS together to see how many A+C+ cells there are.

Response 5- A summary diagram of the number of ALDHhighCD133+ve Ishikawa cells is included in supplementary figure 1.

 References

1.         Kitson S, Maskell Z, Sivalingam VN, Allen JL, Ali S, Burns S, et al. PRE-surgical Metformin In Uterine Malignancy (PREMIUM): a multi-center, randomized double-blind, placebo-controlled phase 3 trial. Clinical cancer research : an official journal of the American Association for Cancer Research. 2018.

2.         Scheen AJ. Clinical pharmacokinetics of metformin. Clinical pharmacokinetics. 1996;30:359-71.

Round 2

Reviewer 2 Report

This is a revision did not address several of the major issues raised in the first review.

1.        The authors respond to concerns about CSC my stating the work cant be done due to time limitations.  While I understand this is a time-consuming assays, I believe it is essential to the paper otherwise the entire premise of the paper fall to the wayside—one cannot comment on the impact of metformin on CSC if CSC are not appropriately defined.  I will let the editors decide, but in my opinion, the paper shouldn’t be published w/o this.  It would be preferable if this were with patient samples, but cell lines would be sufficient.

2.       The response regarding the IHC of the PREMIUM study is also not sufficient.  

i.                     First – the antibody used should be listed in the methods of the paper and not buried in the supplemental where it is hard to find. In addition they need to detail if ALDH and CD133 expression in epithelial compartment only was analyzed – many studies analyze stomal ALDH and this is likely irrelevant.

ii.                   Further, the authors claim this antibody should react against all ALDH1 proteins, but this is not known/or defined.  The antibody used was generated to recognizes ALDH1A1 and it is not known if it reacts to other ALDH1A family members.  The authors should include western blots/IHC vs cell lines with known ALDH1A or ALDH1A3 expression to determine reactivity against each protein and report this.  In the absence of this the authors can only conclude metformin does not impact ALDH1A1 expression.

iii.                 Finally, representative IHC for the ALDH and CD133 should be shown so the reader can evaluate the quality of the IHC.

Author Response

Comment 1: The authors respond to concerns about CSC my stating the work cant be done due to time limitations.  While I understand this is a time-consuming assays, I believe it is essential to the paper otherwise the entire premise of the paper fall to the wayside—one cannot comment on the impact of metformin on CSC if CSC are not appropriately defined.  I will let the editors decide, but in my opinion, the paper shouldn’t be published w/o this.  It would be preferable if this were with patient samples, but cell lines would be sufficient.

Response 1: As the reviewer understands, the experimental work required to address point one would take at least six months to complete. Whilst we agree with the reviewer that this is the ‘gold standard’ method of determining cancer stem cell activity, we believe that previously published work demonstrating that ALDHhigh and CD133+ve cells form tumours in animal xenograft models of endometrial cancer at much lower concentrations than bulk tumour cells supports their use as markers of cancer stem cell activity (1,2). The formation of three dimensional ‘spheres’ under adhesion free conditions has proven to be a reliable in vitro assay to assess cancer stem cell activity (3). We therefore feel that using these validated experimental techniques we have been able to adequately demonstrate an effect of metformin on endometrial cancer stem cell activity. These findings were confirmed in our RT-qPCR experiments, in which metformin reduced the expression of genes associated with self-renewal, pluripotency and a cancer stem cell phenotype. We have included in the discussion on page 10 that there will be a need for in vivo confirmation of our findings and we will defer to the reviewer and editor’s decision as to whether we have provided sufficient evidence at the present time. 

Comment 2: The response regarding the IHC of the PREMIUM study is also not sufficient. 

i. First – the antibody used should be listed in the methods of the paper and not buried in the supplemental where it is hard to find. In addition they need to detail if ALDH and CD133 expression in epithelial compartment only was analyzed – many studies analyze stomal ALDH and this is likely irrelevant.

ii. Further, the authors claim this antibody should react against all ALDH1 proteins, but this is not known/or defined.  The antibody used was generated to recognizes ALDH1A1 and it is not known if it reacts to other ALDH1A family members.  The authors should include western blots/IHC vs cell lines with known ALDH1A or ALDH1A3 expression to determine reactivity against each protein and report this.  In the absence of this the authors can only conclude metformin does not impact ALDH1A1 expression.

iii. Finally, representative IHC for the ALDH and CD133 should be shown so the reader can evaluate the quality of the IHC.

Response 2: We thank the reviewer for their comment.

i. Details of the antibodies used for the IHC experiments have been added to the methods section of the paper on page 13. Only staining in endometrial cancer glands was analysed. This has now been made clearer in the manuscript “…The percentage of cells with positive apical membrane (CD133) or cytoplasmic (ALDH) staining was scored, regardless of staining intensity, in all endometrial cancer glands contained within triplicate repeat cores. Stromal staining was not scored…”

ii. We unfortunately do not have our own Western blot or IHC data to demonstrate the reactivity of the ALDH1 antibody against ALDH1A1 and ALDH1A3 individually, but have included a reference that demonstrates this on page 13.

iii. Representative IHC images have been added to Supplementary Figure 6.

References

1. Rahadiani N, Ikeda J, Mamat S, Matsuzaki S, Ueda Y, Umehara R, et al. Expression of aldehyde dehydrogenase 1 (ALDH1) in endometrioid adenocarcinoma and its clinical implications. Cancer science 2011;102:903-8

2. Nakamura M, Kyo S, Zhang B, Zhang X, Mizumoto Y, Takakura M, et al. Prognostic impact of CD133 expression as a tumor-initiating cell marker in endometrial cancer. Human pathology 2010;41:1516-29

3. Shaw FL, Harrison H, Spence K, Ablett MP, Simoes BM, Farnie G, et al. A detailed mammosphere assay protocol for the quantification of breast stem cell activity. Journal of mammary gland biology and neoplasia 2012;17:111-7

Round 3

Reviewer 2 Report

The IHC presented from the Premium study, the analysis of which is an important part of the study, are extremely poor.  The ALDH1A stain is drastically overstained making it nearly uninterpretable.  If all samples look like this, then of it is no surprise no differences were seen given the level of background stain.  The CD133 stain is also hard to interpret at this resolution (real vs. edge artifact).  

Given this antibody has unclear specificity (likely ALDH1A1 only, but unclear) and the stains are so poor, I think this aspect of the paper should (i) either be repeated with an established ALDH ab and positive stain confirmed by a pathologist, or (ii) be removed before publishing. 

Author Response

We thank the reviewer for their comment. The immunohistochemical data has now been removed from the manuscript and the discussion amended to reflect the concerns raised by the reviewer on line 388-398.